# Pharmacological Inhibition of Gasdermin D Suppresses Angiotensin II-Induced Experimental Abdominal Aortic Aneurysms

**DOI:** 10.3390/biom13060899

**Published:** 2023-05-28

**Authors:** Jia Guo, Jinyun Shi, Min Qin, Yan Wang, Zhidong Li, Takahiro Shoji, Toru Ikezoe, Yingbin Ge, Baohui Xu

**Affiliations:** 1Center for Hypertension Care, Shanxi Medical University First Hospital, Taiyuan 030001, China; shijy0306@sxmu.edu.cn (J.S.); tanmin@sxmu.edu.cn (M.Q.); 2Department of Surgery, Stanford University School of Medicine, Stanford, CA 94305, USA; tshoji@saichu.jp (T.S.); ikezoe@ks.kyorin-u.ac.jp (T.I.); baohuixu@stanford.edu (B.X.); 3Institute of Medical Innovation and Research, Peking University Third Hospital, Beijing 100191, China; yanwang2019@bjmu.edu.cn; 4Department of Pharmacology, Shanxi Medical University, Taiyuan 030001, China; lee2991@sxmu.edu.cn; 5Department of Physiology, Nanjing Medical University, Nanjing 211166, China; ybge@njmu.edu.cn

**Keywords:** abdominal aortic aneurysms, gasdermin D, disulfiram, interleukin-1β, macrophages

## Abstract

Background: Gasdermin D, a molecule downstream of the nucleotide-binding oligomerization domain-like receptor family pyrin domain containing inflammasome, forms the membrane pore for the secretion of interleukin (IL)-1β and IL-18, and also mediates pyroptosis. This study was to explore the influence of treatment with disulfiram, a small molecule inhibitor to gasdermin D, on the formation and progression of experimental abdominal aortic aneurysms (AAA). Methods: AAAs were induced in 10-week-old male apolipoprotein E deficient mice by subcutaneous infusion of angiotensin II (1000 ng/min/kg body weight) for 28 days via osmotic minipumps. Three days prior to angiotensin II infusion, disulfiram (50 mg/kg) or an equal volume of saline as the vehicle control was administered daily via oral gavage. The influence on experimental AAAs was analyzed by serial measurements of aortic diameters via ultrasonography, grading AAA severity and histopathology at sacrifice. Serum IL-1β and IL-18 levels, systolic blood pressure, total cholesterol, and triglyceride were also measured. Additional experiments assayed the influences on the cell viability and IL-1β secretion of in vitro activated macrophages. Results: Disulfiram significantly reduced the enlargement, incidence, and severity of angiotensin II-induced experimental AAAs with attenuation of medial elastin breaks, mural macrophage accumulation, and systolic blood pressure. The AAA suppression was also associated with reduced systemic levels of IL-1β but not IL-18. However, disulfiram treatment had no impact on body weight gain and lipid levels in aneurysmal mice. Additionally, disulfiram treatment also markedly reduced the secretion of IL-1β from activated macrophages with a limited effect on cell viability in vitro. Conclusions: Gasdermin D inhibition by disulfiram attenuated angiotensin II-induced experimental AAAs with reduced systemic IL-1β levels and in vitro activated macrophage IL-1β secretion. Our study suggests that pharmacological gasdermin D inhibition may have translational potential for limiting clinical AAA progression.

## 1. Introduction

Abdominal aortic aneurysm (AAA) is a life-threatening chronic disease characterized by progressive dilation of the abdominal aorta, particularly the infrarenal aorta, to a diameter of more than 3.0 cm [1]. AAAs are generally asymptomatic and identified accidentally or through an AAA screening program. Advanced age, male sex, cigarette smoking, white race, and cardiovascular morbidity increase the risk of AAA, whereas female sex, black race, and diabetes reduce the risk of AAA [2]. Surgical (open and endovascular) repair remains the sole treatment for AAA disease. However, only a small portion of AAA patients meet current elective surgical repair criteria (aneurysm diameters of >5.5 cm and 4.5 cm in men and women, respectively) [3,4], whereas a large number of patients with an aneurysm diameter below the surgical repair threshold are at risk of unpredicted lethal rupture. Therefore, innovating nonsurgical pharmacological therapeutic options are needed to limit small AAA progression and ultimately to reduce mortality due to unpredicted premature AAA rupture in patients who do not meet surgical criteria [5].

Inflammation critically mediates AAA pathogenesis by aortic leukocyte accumulation, proteolysis via matrix metalloproteinases, imbalanced pro-and anti-inflammatory cytokines, elevated renin-angiotensin activity, and pathogenic mural angiogenesis [6,7,8,9,10,11]. For example, genetic or pharmacological inhibition of IL-1β and its receptor suppressed experimental AAAs induced by the intra-aortic infusion or aortic wall topical painting of porcine pancreatic elastase as well as Kawasaki disease-associated experimental AAAs [12,13,14,15,16,17]. Genetic ablation of IL-18 or its receptor also mitigated angiotensin II (Ang II)-induced AAAs in apolipoprotein E (ApoE) deficient mice [18,19]. Inflammasomes are a large multimolecular complexes and are critical for inflammatory responses [20]. Inflammasomes are activated to generate active caspases, which cleave the precursors of interleukin (IL)-1β and IL-18 to the corresponding functional mature forms [20,21]. The nucleotide-binding oligomerization domain-like receptor family pyrin domain containing 3 (NLRP3) inflammasome is the best characterized [22,23]. Almost, if not all, genetically or pharmacologically targeted individual NLRP3 components have been shown to suppress or even prevent experimental AAAs in distinct rodent AAA models [13,16,17,24,25].

Gasdermin D is a molecule downstream of the NLRP3 inflammasome [26,27,28]. Following NLRP3 inflammasome activation, active caspase 1 cleaves gasdermin D into the C- and N-terminals, the latter forming cell membrane pores for the secretion of mature proinflammatory IL-1β and IL-18, whereas excessive pore formation by cleaved gasdermin D leads to cell death (known as pyroptosis) [26,27,29,30,31,32]. Recent studies have shown the pathogenic importance of gasdermin D in the pathogenesis of several vascular disorders. For example, genetic deficiency or AAV5-mediated suppression of gasdermin D attenuated experimental atherosclerosis disorders [33,34,35,36]. Gasdermin D deficiency attenuated experimental ischemic stroke [37,38,39]. Cardiomyocyte-specific gasdermin D ablation and siRNA gasdermin suppression, respectively, also abrogated transverse aortic construction- and Ang II-induced cardiac hypertrophy disorders [40,41]. Additionally, gasdermin D deficiency also attenuated myocardial reperfusion injury in mice [38]. However, the role that gasdermin D plays in AAAs has not been well investigated.

Given the importance of NLPR3 in AAA pathogenesis and gasdermin D as an executive molecule in inflammasomes, we hypothesized that blocking gasdermin D activity inhibits experimental AAAs. A drug in clinical use, disulfiram, has been shown to block the pore formation mediated by the cleaved gasdermin D N-terminal and thus prevents the release of IL-1β and IL-18 [42]. Thus, the present study investigated the influence of the gasdermin D inhibitor disulfiram on Ang II-induced experimental AAAs via noninvasive ultrasound imaging, histopathology, cytokine assays, and macrophages in vitro. We found that gasdermin D inhibition by disulfiram effectively attenuated experimental AAA formation and progression in association with reduced systemic and macrophage-derived IL-1β levels without an effect on lipid levels.

## 2. Materials and Methods

### 2.1. Induction and Treatment of Experimental AAAs

Ten- to twelve-week-old male ApoE deficient mice on C57BL/6 genetic background were purchased from Nanjing Junke Bioengineering Ltd., Nanjing, Jiangsu, China and used throughout this study. The use and care of experimental animals in this study were approved by Shanxi Medical University Animal Research Committee and complied with the University Laboratory Animal Research Guidelines (Taiyuan, Shanxi, China). Mice were housed at Shanxi Medical University Research Animal Facility with free access to standard mouse chow and drinking water at a constant temperature of 22 ± 2 °C and humidity of 60–65% with a 12 h dark/light cycle. Mice were monitored daily, and body weight was measured weekly.

AAAs were induced by subcutaneous infusion of Ang II, as previously reported [8,11,43,44]. Briefly, mice were anesthetized by inhalation of 3% isoflurane and oxygen. Osmotic pumps (Model 2004, Durect Corporation, CA, USA) were surgically implanted into the dorsal subcutaneous pocket to deliver human recombinant Ang II (1000 ng/kg/min, catalog number ab120183, abcam, Waltham, MA, USA) for 28 days. Three days prior to Ang II infusion, gasdermin D inhibitor disulfiram (50 mg/kg/day, catalog number A0322A, Meilunbio, Dalian, Liaoning, China) or an equal value of vehicle (saline) was orally given to the mice for a total of 30 days. Disulfiram at 50 mg/kg/day, effectively suppressing gasdermin D activity in the mouse sepsis model, was within the clinical dose range (500 mg and 250 mg for initiating and subsequently maintaining dose, respectively, for people with a body weight of 60 kg) at allometric scale (http://clymer.altervista.org/minor/allometry.html (accessed on 15 May 2023) [42].

### 2.2. Ultrasound Imaging of Experimental AAAs In Vivo

Experimental AAAs were monitored by serial measurements of maximal suprarenal aortic luminal diameters using the S-sharp prospect 3.0 small animal ultrasound system (S-Sharp Corporation, Nepa Gene Co., Ltd., Ichikawa, Chiba, Japan). Time points for diameter measurements included the baseline level (day 0) prior to and days 7, 14, 21, and 28 following Ang II infusion [8,11]. All measurements were conducted by two investigators blinded to treatment group assignment. An AAA was defined by the presence of aortic dissection imaged via ultrasonography, a 50% increase in aortic diameter over the baseline, or death due to AAA rupture [8,11].

### 2.3. Measurement of Blood Pressure

Blood pressure was determined using a noninvasive tail-cuff monitoring system, Mode BP-2010A, Softron Biotechnology, Haidian, Beijing, China [43]. The mouse was placed in a temperature-controlled restrainer. Fifteen minutes thereafter, blood pressure and heart rate were recorded for five consecutive measurements and averaged. All measurements were performed between 8.0 am and 10.0 am for the baseline level and day 28 after Ang II infusion.

### 2.4. AAA Severity Grading

Twenty-eight days after Ang II infusion, mice were sacrificed by overdose isoflurane inhalation, perfused with phosphate-buffered saline via the left ventricle. The entire aorta was then isolated, and AAA severity in suprarenal aorta was scored as grade 0 to grade V, as previously reported [11,45]. Individual grades were grade 0: no recognizable aortic dilation; grade I: recognizable aortic dilation without thrombus; grade II: aortic dilation with recognizable thrombus; grade III: pronounced bulbous form of aneurysm with thrombus; grade IV: multiple AAAs with thrombus; and grade V: death due to AAA rupture.

### 2.5. Determination of Serum Triglycerides and Total Cholesterol Levels

Mice were anesthetized by isoflurane inhalation after the last aortic diameter measurement. Peripheral blood was obtained by retro-orbital bleeding. The serum was prepared by centrifugation at 3000 rpm for 30 min. Serum specimens were isolated and stored at −20 °C until analysis. Serum lipid profiles including total cholesterol and triglyceride were measured using an automatic chemistry analyzer, Servicebio, Wuhan, Hubei, China.

### 2.6. Aortic Histological Analyses

Twenty-eight days following Ang II infusion, mice were sacrificed by overdose inhalation of isoflurane. Suprarenal aorta was dissected, fixed with 4% paraformaldehyde, embedded in paraffin, and sectioned (6 mm) for all histological analyses. Medial elastin laminae were assessed via Elastica van Gieson (EVG) stain [8,46]. Following deparaffinization, rehydration, and antigen retiral, sections were stained with an antibody against CD68 (macrophages) using the standard immunohistochemistry procedure and 3,3′-diaminobenzidine as the peroxidase substrate [10,47,48]. Immunostained sections were further counterstained with hematoxylin, mounted, and coverslipped. Images for all staining were acquired on an Olympus microscope equipped with a digital camera.

### 2.7. Culture and Treatment of Macrophages In Vitro

Mouse macrophage cell line RAW264.7 was obtained from the American Type Culture Collection, Manassas, VA, USA. Macrophages were cultured in Dulbecco’s Modified Eagle’s Medium supplemented with 10% fetal bovine serum and 1% penicillin and streptomycin (Thermo Fisher Scientific, Waltham, MA, USA) at 37 °C and 5% CO_2_ and used within 10 passages. Cells were activated with lipopolysaccharide (100 ng/mL, Cat#: L2880, Sigma Aldrich, St. Louis, MO, USA) for 24 h followed by additional 24 h incubation with vehicle, Ang II (100 nM) and/or disulfiram (25 mM) [11,42,49]. At the end of the experiments, cells were subjected to viability assessment, and culture supernatants were harvested to determine IL-1b release.

### 2.8. Enzyme-Linked Immunosorbent Assays (ELISA)

ELISA assays were used to determine cytokine levels in serum, culture cell supernatants, and cell viability using commercially available kits and performed according to individual manufacturer’s instructions. These were ELISA kits for mouse IL-1β and IL1-8 (catalog # JL18442 and JL20253, J&L Biological, Baoshan, Shanghai, China) and CCK8 cell counting (Dojindo Laboratories, Chaoyang, Beijing, China).

### 2.9. Statistical Analysis

All statistical analyses were performed with GraphPad Software Prism version 9.5.1, San Diego, CA, USA. Normality and outlier data for all continuous variables were evaluated using ROUT and Shapiro–Wilk tests, respectively. Data for continuous variables with normal distribution were thus presented as mean and standard deviation. Two-way ANOVA followed by two group comparison or Student’s *t*-test was used to test the difference for normally distributed data. All non-normally distributed data were presented as media and interquartile (25th and 75th), and statistical difference was tested using a nonparametric Mann–Whitney test. Chi-square test was employed to test the difference for the AAA severity score distribution between two treatment groups. The difference in cumulative AAA incidence and mortality between the two groups was tested using the log-rank test. A *p* value of less than 0.05 was considered statistically significant.

## 3. Results

### 3.1. Disulfiram Treatment Suppresses the Formation and Progression of Experimental AAAs

To evaluate the influence of gasdermin D inhibition on experimental AAAs, Ang II was subcutaneously delivered to male Apo E^−/−^ mice (1000 ng/min/kg body weight) for 28 days. Gasdermin D inhibitor disulfiram (50 mg/kg) or an equal value of vehicle (saline) was administrated to the mice via oral gavage for 30 days. Maximal suprarenal aortic diameters were measured using a small animal ultrasonography prior to and weekly after Ang II infusion. As shown in Figure 1A,B, no difference was observed in the baseline suprarenal aortic diameter between the vehicle (mean and SD: 1.03 ± 0.04 mm) and disulfiram (1.01 ± 0.04 mm) treated groups. In vehicle-treated mice, Ang II infusion induced a time-dependent expansion in the suprarenal aortic diameter as compared to the baseline level. In contrast, disulfiram treatment dramatically attenuated Ang II infusion-induced aortic diameter enlargement. Aortic diameters were significantly smaller in disulfiram- than those in vehicle-, treated mice on day 21 (1.47 ± 0.04 mm and 1.22 ± 0.04 mm for vehicle and disulfiram treatments, respectively, *p* < 0.01) and day 28 (1.58 ± 0.04 mm and 1.25 ± 0.04 mm for vehicle and disulfiram, respectively, *p* < 0.01) after Ang II infusion.

AAAs, defined by the presence of aortic dissection noticed by ultrasonography, a 50% or more increase in aortic diameter over the baseline, or death due to AAA rupture, were observed in 10 out of 13 mice (77%) within 28 days following Ang II infusion (Figure 1C). In contrast, AAAs were noted in only 2 out of 13 mice (15%) in disulfiram-treated mice. However, there was no difference in mortality due to AAA rupture between the two treatment groups (Figure 1D). One mouse died accidently in the disulfiram treatment group. Death due to AAA rupture was noted in two mice (days 3 and 6) and one mouse (day 18) in the vehicle and disulfiram treatment groups, respectively, following Ang II infusion. Altogether, these results indicate that pharmacological inhibition of gasdermin D by disulfiram suppresses experimental AAA formation and progression.

### 3.2. Disulfiram Treatment Ameliorates Experimental AAA Severity

At sacrifice, we further assessed the influence of disulfiram treatment on AAA severity as a complementary approach to ultrasound imaging. AAA severity was scored according to the presence or absence of aortic dilation and thrombus as well as the shape and number of AAAs. As shown in Figure 2A,B, AAAs were scored as grade V (death due to AAA rupture) in two mice, grade IV in one mouse, grade II in three mice, grade II in five mice, and grade I in two mice in the vehicle treatment group. In contrast, AAAs were scored as grade I in eleven mice, grade II in one mouse, and grade V in one mouse in the disulfiram treatment group. Additionally, the average AAA severity score was significantly lower in disulfiram- than in vehicle-treated mice (Figure 2C). Thus, inhibiting gasdermin D by disulfiram reduces experimental AAA severity.

### 3.3. Disulfiram Treatment Attenuates Aortic Inflammation in Experimental AAAs

Further, histopathological analyses were performed to evaluate the influence of disulfiram treatment on key aneurysmal pathologies (Figure 3). As illustrated in H&E stain, severe inflammation was seen in the vehicle- but not in the disulfiram-treated Ang II-infused mice. Medial elastin breaks were readily observed in aortas of the vehicle- as compared to the disulfiram-treated mice. Similarly, macrophages, as indicated by CD68 antibody staining, were apparently noted in aortas from vehicle-treated mice. In contrast, macrophages were rarely seen in aortas from disulfiram-treated mice. Thus, inhibition of gasdermin D by disulfiram suppressed experimental AAAs in conjunction with attenuated aortic inflammation.

### 3.4. Disulfiram Treatment Reduces the Systemic Levels of IL-1β Not IL-18 in Experimental AAAs

Cleavage of gasdermin D by active caspase 1 generates N-terminal, which forms the cell membrane core required for the section of mature IL-1β and IL-18. Given the importance of IL-1β and IL-18 in AAA pathogenesis, we examined whether attenuation of experimental AAAs by gasdermin D inhibitor disulfiram is associated with reduced systemic levels of IL-1β and IL-18. As seen in Figure 4A, serum IL-1β levels were markedly and significantly reduced in disulfiram- as compared to vehicle-treated Ang II-infused mice (*p* < 0.01). Serum IL-1β levels were 547.7 ± 231.8 pg/mL and 272.1 ± 128.0 pg/mL in vehicle and disulfiram treatment groups, respectively, representing 50% reduction following disulfiram treatment. Six and eight mice among twelve and nine mice of vehicle- and disulfiram-treated mice, respectively, had IL-1β levels below the mean levels seen in vehicle treatment group. However, serum IL-18 levels did not differ between vehicle (83.9 ± 43.0 pg/mL) and disulfiram (87.4 ± 58.3 pg/mL) treatment groups. Similarly, 33% and 40% of mice in the vehicle and disulfiram treatment groups, respectively, had serum IL-18 levels over the mean IL-18 level noted in the vehicle treatment group (Figure 4B). These results suggest that reduction of systemic IL-1β levels may contribute to disulfiram-mediated experimental AAA attenuation.

### 3.5. Disulfiram Treatment Slightly Lowers Angiotensin II-Induced Increase in Systolic Blood Pressure in Experimental AAAs

Systolic blood pressure was measured at the baseline levels as well as 28 days after Ang II infusion via the noninvasive tail cuff method (Figure 5A). No difference in baseline systolic blood pressure was found between the vehicle (105.6 ± 6.4 mm Hg) and disulfiram (105.7 ± 8.4 mm Hg) treatment groups (Figure 5A). Following subcutaneous Ang II infusion for 28 days, the systolic blood pressure was significantly elevated in both treatment groups as compared to the corresponding baseline level (*p* < 0.05 and *p* < 0.01 for disulfiram and vehicle treatments, respectively). Unexpectedly, systolic blood pressure measured on day 28 was slightly but significantly lower in the disulfiram (123.9 ± 4.9 mm Hg) than that in the vehicle (144.6 ± 6.5 mm Hg)-treated group (*p* < 0.05). Additionally, Ang II infusion (450.3 ± 66.1 and 452.0 ± 79.5 beats/minute for vehicle and disulfiram treatments, respectively) also mildly but significantly reduced heart rate as compared to respective baseline levels (538.5 ± 58.1 and 557.2 ± 30.1 beats/minute for vehicle and disulfiram treatments, respectively) (*p* < 0.01) regardless of treatment with vehicle or disulfiram (Figure 5B). Heart rate on day 28, however, did not differ between vehicle and disulfiram treatment groups.

### 3.6. Disulfiram Treatment Affects Neither Body Weight Gain Nor Lipid Levels in Experimental AAAs

We assessed whether the AAA suppression by disulfiram is associated with alterations in body weight gain and lipid levels in experimental AAA mice. As shown in Figure 5C, there was no difference in baseline body weight between the vehicle (21.9 ± 1.3 g) and disulfiram (22.1 ± 1.2 g) treatment groups. Disulfiram treatment did not substantially reduce the gain of body weight, which were measured as 25.2 ± 1.3 g and 24.6 ± 1.4 g for vehicle and disulfiram treatments, respectively, 28 days after Ang II infusion. Additionally, no difference was found in serum total cholesterol levels between vehicle (627.6 ± 17.8 mg/dL) and disulfiram (597.9 ± 18.2 mg/dL) treatment groups (Figure 5D,E). Similarly, disulfiram treatment did not impact serum triglyceride as compared to vehicle treatment (114.9 ± 3.3 mg/dL and 106.2 ± 4.0 mg/dL in vehicle and disulfiram treatments, respectively) (Figure 5D,E). Altogether, these results suggest that the suppression of experimental AAAs by disulfiram treatment was independent of body weight gain or hyperlipidemic conditions.

### 3.7. Disulfiram Treatment Suppresses IL-1β Release by Macrophages In Vitro

Gasdermin D acts as a pore-forming protein and contributes to inflammasome-mediated secretion of IL-1β and IL-18 as well as cellular pyroptosis. Macrophages outnumber other inflammatory cells in an aneurysmal aorta and critically mediate AAA pathogenesis via the production of cytokines and other mediators. We therefore tested whether disulfiram influences macrophage viability and IL-1β secretion following in vitro LPS-mediated activation. In CCK8 assays (Figure 6A), disulfiram-treated macrophage viability was 82% (SD: 0.0%) of that seen in vehicle treatment (theoretical value: 100%) in the absence of Ang II following LPS-mediated activation. Such an effect was, however, not observed in the presence of Ang II (93.3 ± 17.6% of vehicle treatment). These data suggest that disulfiram had limited effect on macrophage viability.

As shown in Figure 6B, supernatant IL-1β concentration, as measured by ELISA, was almost identical in the absence (564.7 ± 25.7 pg/mL) or presence (562.0 ± 34.4 pg/mL) of Ang II in LPS-activated macrophages. In contrast, disulfiram treatment significantly reduced supernatant IL-1β concentration regardless of the presence or absence of Ang II (*p* < 0.01 compared to vehicle treatment). Although IL-1β concentration was lower in cotreatment with disulfiram and Ang II (310.0 ± 11.1 pg/mL) than that in treatment with disulfiram alone (384.0 ± 57.3 pg/mL), this difference was not statistically significant. These results confirmed that gasdermin D inhibitor disulfiram suppressed IL-1β secretion by macrophages.

## 4. Discussion

In the present study, treatment with gasdermin inhibitor disulfiram reduced aneurysmal aortic diameter enlargement, lowered AAA incidence, and ameliorated aneurysm severity in ApoE deficient mice following Ang II infusion. Disulfiram treatment also reduced systemic (serum) levels of IL-1β but not IL-18 in vivo as well as IL-1β release from LPS-activated macrophages in vitro. While disulfiram treatment slightly lowered systolic blood pressure, no remarkable influences were noted for hypercholesterolemia and body weight gain. Nevertheless, our findings indicate that gasdermin D inhibition by disulfiram attenuates Ang II-induced experimental AAAs.

Two recent studies have suggested the importance of gasdermin D in AAA pathogenesis using genetically modified animals. For instance, hyperlipidemic mice deficient for gasdermin D were protected against aortic rupture resulting from Ang II infusion (hyperlipidemia induced by inoculating adeno-associated virus encoding a human proprotein convertase subtilisin/kexin type 9 gain-of-function of mutation (D377Y) or long-term feeding lysyl oxidase inhibitor β-aminopropionitrile [50]). Smooth muscle cell-specific deletion of gasdermin D also attenuated aneurysm diameter expansion and reduced AAA incidence in experimental AAAs created by subcutaneous Ang II infusion or topical application of elastase to aortic adventitia [51]. However, AAA suppression caused by smooth muscle cell-specific gasdermin D deficiency modified the phenotype of smooth muscle cells due to reduced endoplasmic reticulum stress and production of the ornithine decarboxylase-catalyzed metabolite putrescine, but not accompanied by smooth muscle cell pyroptosis [51]. Altogether, these previous studies suggest that gasdermin D may be a potential therapeutic target for treating AAA disease.

Highly similar to the experimental construct utilized in the present study, disulfiram treatment has been shown to reduce aneurysmal diameter, aortic collagen, and elastin fragmentation in ApoE deficient mice following Ang II infusion [52]. Unexpectedly, disulfiram treatment also downregulated Ang II-induced protein expression for NLRP3, ACS, caspase 1, IL-1β, IL-18, and gasdermin D in smooth muscle cells in vitro, which is not consistent with the hypothesis that disulfiram inhibits gasdermin D activity by interrupting pore formation [42]. No data were provided on whether disulfiram treatment influences active/cleaved levels of caspase 1, IL-1β, IL-18, and gasdermin D or directly reduces smooth muscle cell pyroptosis (either functional activity and imaging excessive pore formation). Additionally, it was also unclear from this previous study whether disulfiram treatment affected systemic levels and the ability of macrophages to secrete, IL-1β and IL-18. Despite these limitations, the authors concluded that disulfiram suppressed experimental AAAs by ameliorating smooth muscle cell pyroptosis. In addition to the attenuation of AAA progression and severity by disulfiram, the present study provides complementary evidence that disulfiram treatment reduced the systemic (serum) levels of IL-1β, LPS-stimulated macrophage IL-1β secretion, and slightly increased activated macrophage death. Nevertheless, the present study together with the previous study support the conclusion that pharmacological inhibition of gasdermin D attenuates experimental AAAs.

Gasdermin D consists of a 31 kDa N-terminus and 22 kDa C-terminus linked by a peptide. Active caspase 1 and caspases 4, 5, and 11 in canonical and noncanonical pathways, respectively, specifically cleave the gasdermin D linker at D275 [21,53,54]. The resultant N-terminus forms approximately 27 monomers and assembles into a transmembrane pore with an inner diameter of 18 nm, which is either required for the release of active forms of IL-1β and IL-18, or leads to host cell lysis (known as pyroptosis) [55,56]. Genetic deficiency, or pharmacological inhibition of IL-1β, IL-18 or their receptors, has been demonstrated to suppress experimental AAAs in distinct AAA models [12,13,14,15,16,18]. In accordance with the pathogenic significance of IL-1β in AAAs, systemic IL-1β levels, as measured by serum IL-1β, were significantly reduced in disulfiram- as compared to vehicle-treated Ang II-infused ApoE deficient mice. While average serum IL-18 levels were slightly decreased, no statistical difference was found between the two treatment groups. These results suggest that reduced systemic IL-1β levels may be partially attributed to impaired release due to disulfiram-attenuated gasdermin D activity. In our further in vitro experiments, disulfiram treatment markedly reduced IL-1β secretion by LPS-activated mouse macrophages while slightly reducing cell viability as measured by CCK8 assay. Consistent with our findings, disulfiram has been shown to attenuate IL-1β secretion by acute monocytic leukemia patient-derived human monocytes following LPS priming in vitro [42,57,58]. Altogether, our findings suggest that AAA suppression by the gasdermin D inhibitor disulfiram may be mediated largely by damping macrophage IL-1β secretion with a minor contribution from macrophage pyroptosis.

Disulfiram treatment slightly but significantly lowered systolic blood pressure in Ang II-infused, ApoE deficient mice. Consistent with the present study, the short- or long-term disulfiram treatment has been reported to slightly or moderately lower systolic blood pressure in people and rodents [59,60,61,62,63,64,65]. Thus, the question is whether reduced blood pressure played a role in AAA attenuation by disulfiram. In epidemiological studies, hypertension or high blood pressure was associated with a mild increase in AAA risk in retrospective but not prospective studies [66,67]. Neither hypertension nor pharmacologically managing blood pressure modulated the clinical AAA enlargement rate [68,69,70,71]. In experimental studies, several classes of blood pressure-lowering drugs, including inhibitors to Ang II type 1 receptor, Ang-converting enzyme, calcium channel, and β-adrenergic receptor, have been shown to suppress experimental AAAs [9,43,72,73,74,75,76,77,78,79,80]. However, lowering blood pressure by endothelin 1 receptor antagonist (bosentan) or hydralazine failed to inhibit experimental AAAs in ApoE deficient mice induced by Ang II infusion [43,81]. Additionally, increasing blood pressure by phenylephrine also did not counteract AAA suppression by calcium channel blocker diltiazem [76]. Based on these published studies, it seems unlikely that the systolic blood pressure reduction observed in our study contributed to disulfiram-mediated AAA suppression.

High levels of total cholesterol, but not triglyceride, are associated with increased risk for clinical AAAs [82,83]. Hypercholesterolemia, obesity, or blocking transforming growth factor β1 activity is required for the successful induction of experimental AAAs by Ang II infusion, but not elastase infusion [44,84,85,86,87]. However, we found that treatment with disulfiram for 30 days did not alter serum levels of cholesterol and triglycerides. Additionally, loss of body weight attenuated AAAs in high-fat diet fed normolipidemic mice following Ang II infusion [88]. In the present study, no significant reduction in body weight was observed in disulfiram- as compared to vehicle-treated Ang II-infused ApoE deficient mice. Thus, AAA suppression by disulfiram treatment was independent of either lipid levels or body weight gain.

Disulfiram (Antabuse) is an aldehyde dehydrogenase inhibitor approved for treating alcohol dependence with well documented clinical safety (https://www.ncbi.nlm.nih.gov/books/NBK459340/ (accessed on 15 May 2023). Many clinical trials have been completed or are ongoing to test the therapeutic efficacy of disulfiram in other medical conditions, including infectious disease, retina degeneration, and malignant tumors (ClinicalTrials.gov (accessed on 15 May 2023). Additional mechanistically distinct small molecule gasdermin D inhibitors have also been reported [89,90,91]. Thus, regardless of the mechanisms by which disulfiram attenuated experimental AAAs, our findings together with those from other recent studies suggest that pharmacologically inhibiting gasdermin D may have the translational application for limiting clinical AAA progression.

We focused on the pharmacological inhibition efficacy of disulfiram treatment on experimental AAAs, thus our study has limitations. First, we did not assess whether gasdermin D was activated in aneurysmal lesion. However, in the study conducted by Gao and colleagues, the cleaved N-terminal of gasdermin D, an indicator for gasdermin D activation, was substantially increased in clinical and experimental AAA specimens [51]. Second, due to the limited aortic tissue available, we were unable to examine whether full gasdermin D was cleaved in aneurysmal aorta following disulfiram treatment. Third, the present study only applied a single AAA model and one clinically relevant disulfiram dose to test the preventive influence on experimental AAAs. For better translational application, further studies should validate whether disulfiram treatment is effective in mitigating the progression of existing AAAs in alternative AAA models with drug dose range design.

The findings that have emerged from the present and previous studies [50,51,52] led us to propose a working model that gasdermin D contributes to experimental AAA pathogenesis (Figure 7). In AAAs, either canonical or noncanonical inflammasome activation cleaves procaspases 1, 4, 5, 7, and 11 to active forms [16,52]. Certain viral proteinases also activate procaspases [92,93]. Then, active caspases cleave gasdermin D to the N- and C-terminals (gasdermin D activation). The N-terminal forms cell membrane pores which either promote the secretion of proaneurysmal IL-1β or release various types of proaneurysmal mediators due to cell death (pyroptosis). Additionally, gasdermin D activation also results in smooth muscle cell (SMC) phenotype switching, namely from contractible towards secretory SMCs that contribute to AAA pathogenesis by producing proinflammatory mediators in a pore-independent ways. Disulfiram, a specific gasdermin D inhibitor, mediates AAA suppression in part by inhibiting gasdermin D cleavage, which consequently diminishes the release of proaneurysmal mediators from macrophages and secretory SMCs in pore-dependent and/or -independent mechanisms.

## 5. Conclusions

Pharmacological inhibition of gasdermin D by disulfiram suppresses angiotensin II-induced experimental AAAs. Our study, together with previous reports, suggest that disulfiram or alternative small molecule inhibitors to gasdermin D may hold translational value for nonsurgical management of clinical AAA enlargement. 

## Figures and Tables

**Figure 1 biomolecules-13-00899-f001:**
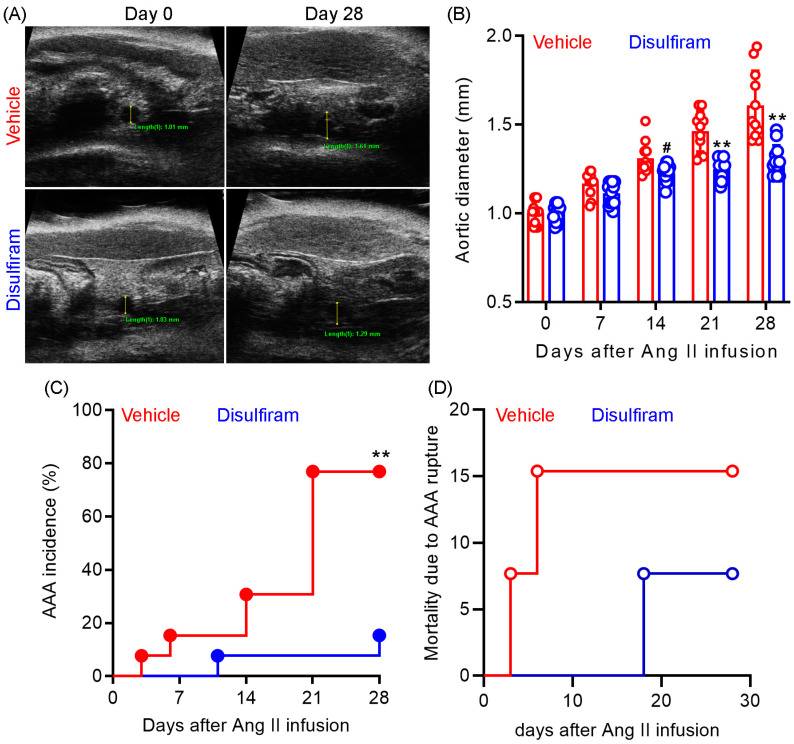
Disulfiram treatment attenuates the formation and progression of experimental AAAs. Experimental AAAs were induced in ApoE deficient male mice by subcutaneously infusing angiotensin II for 28 days. Mice were treated with gasdermin D inhibitor disulfiram or vehicle (saline) and monitored for AAA formation and progression via ultrasonography. (**A**): Representative ultrasound images of suprarenal aortas at the baseline and day 28 in vehicle and disulfiram treatment groups following angiotensin II infusion. (**B**): Mean ± standard deviation of maximal suprarenal aortic diameters at different time points in vehicle- and disulfiram-treated, angiotensin II-infused ApoE deficient mice. Two-way ANOVA followed by two group comparison, 0.05 < ^#^
*p* <0.1 and ** *p* < 0.01 compared to vehicle treatment at same time point. (**C**): AAA incidence. AAA: the presence of aortic dissection imaged by ultrasonography, a 50% or more increase in aortic diameter over the baseline, or death due to AAA rupture. Log-rank test, ** *p* < 0.01 compared to vehicle treatment. (**D**): Mortality due to AAA rupture. *n* = 13 mice/group. Aortic diameters in panel A: 1.01 mm and 1.61 mm on days 0 and 28 for vehicle treatment and 1.03 mm and 1.29 mm on days 0 and 28 for disulfiram treatment.

**Figure 2 biomolecules-13-00899-f002:**
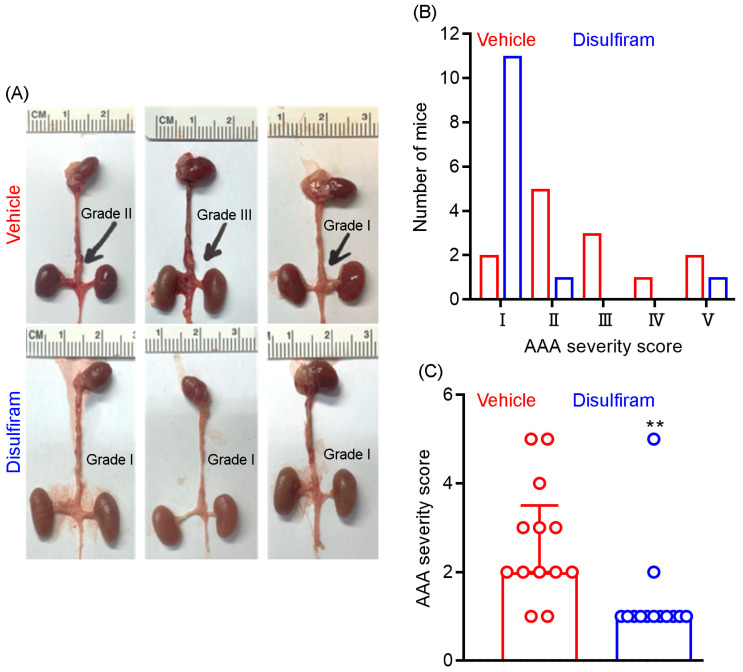
Disulfiram treatment reduces experimental AAA severity. ApoE deficient mice were infused with angiotensin II (1000 ng/kg/min) for 28 days using the subcutaneous implantation of osmotic pumps. Twenty-eight days thereafter, mice were sacrificed, and AAA severity in suprarenal aorta was graded as I to IV based on the presence of aortic dilation, intramural hematoma, and the number of AAAs. An AAA in which the mouse died due to AAA rupture was graded as V. (**A**): Representative macroscopic appearance of aneurysms in the vehicle (graded as II, III, and I from left to right) and disulfiram (all graded as I) treatment groups. (**B**): Distribution of AAA severity scores in vehicle and disulfiram treatment groups. Chi-square test, *p* < 0.01 between two groups. (**C**): Media and interquartile (25th and 75th) of AAA severity in the vehicle and disulfiram treatment groups. Nonparametric Mann–Whitney test, ** *p* < 0.01 compared to vehicle treatment. *n* = 13 mice/group.

**Figure 3 biomolecules-13-00899-f003:**
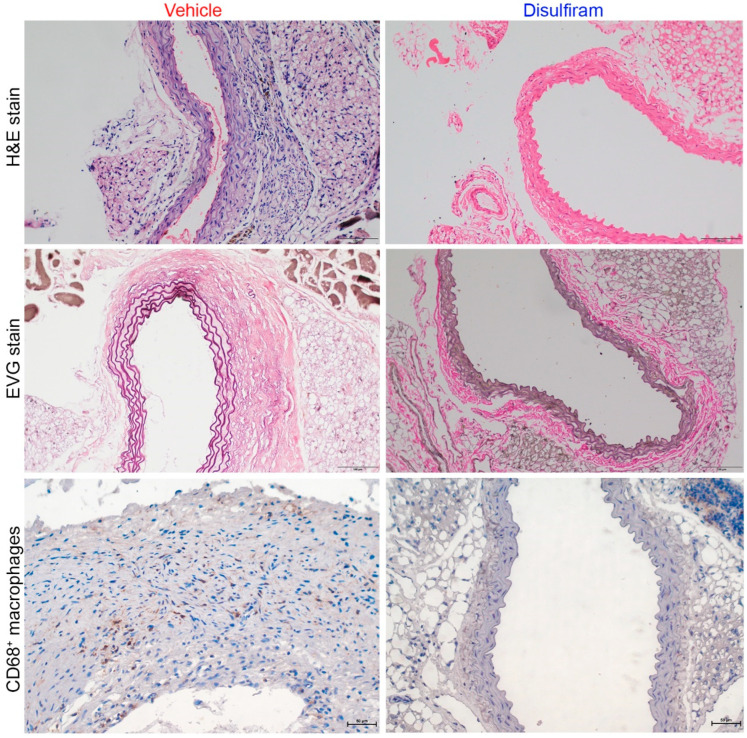
Disulfiram treatment mitigates aneurysmal histopathologies. Histological analyses were performed on paraffin-embedded aortic sections of vehicle- and disulfiram-treated ApoE deficient mice 28 days after angiotensin II infusion. Upper panels (H&E stain): less aortic wall inflammation in disulfiram- as compared to vehicle-treated aneurysmal mice. Middle panels (Elastica van Gieson (EVG) stain): well organized dense medial elastin lamella in disulfiram- as compared to vehicle-treated aneurysmal mice. Lower panels: reduced aortic macrophages in disulfiram- as compared to vehicle-treated aneurysmal mice. Magnification: ×200 for upper and middle panels and ×400 in lower panel. Scale bar: 100 µm in upper and middle panels and 50 µm in lower panels.

**Figure 4 biomolecules-13-00899-f004:**
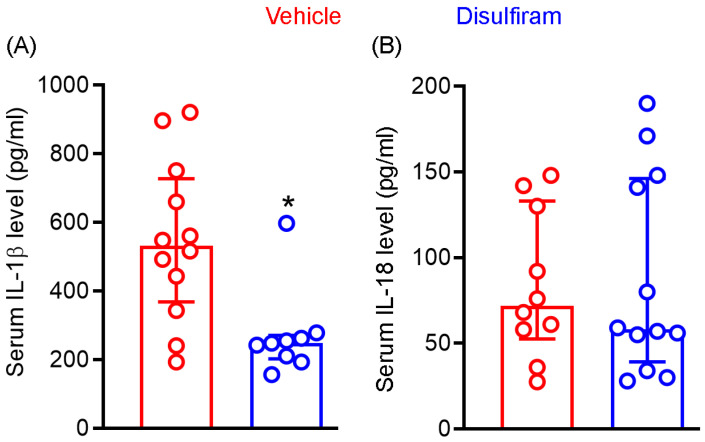
Disulfiram treatment reduces the systemic levels of IL-1β not IL-18 in experimental AAA mice. Twenty-eight days after angiotensin II infusion, sera were prepared from all ApoE^−/−^ mice. IL-1β and IL-18 levels were measured via the ELISA method. (**A**): IL)-1β. (**B**): IL-18. All data are media and interquartile (25th and 75th). Nonparametric Mann–Whitney test, * *p* < 0.05 compared to vehicle treatment group. *n* = 9–12 mice/group.

**Figure 5 biomolecules-13-00899-f005:**
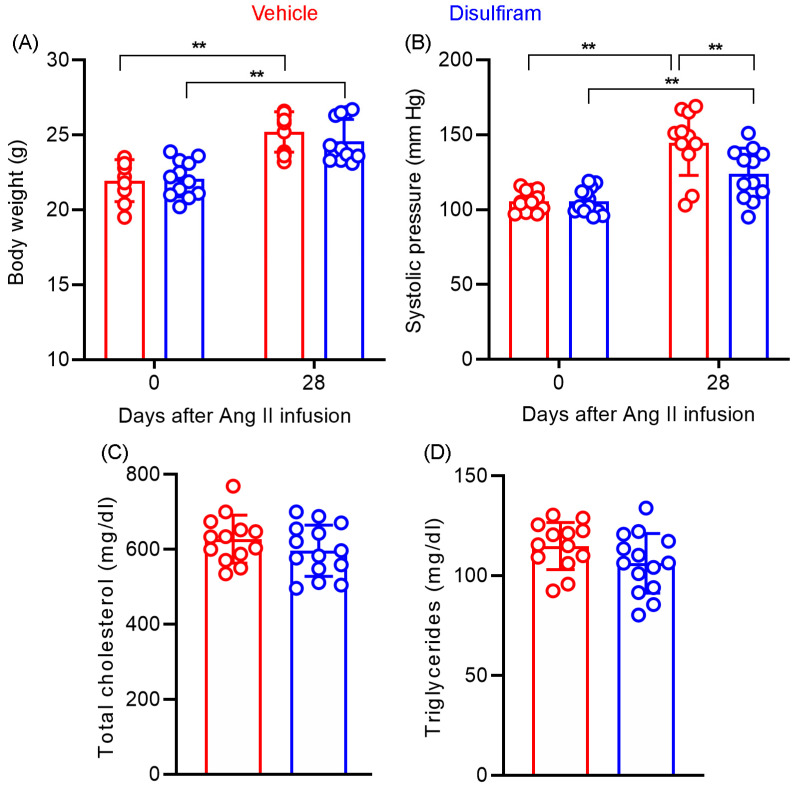
Disulfiram treatment slightly lowers systolic blood pressure without affecting body weight and lipid levels in ApoE^−/−^ mice 28 days following angiotensin II infusion. (**A**): Body weight at the baseline and 28 days after angiotensin II infusion. (**B**): Systolic blood pressure at the baseline and 28 days after angiotensin II infusion measured via the tail-cuff method. (**C**,**D**): Total serum cholesterol (**C**) and triglyceride (**D**) levels in ApoE^−/−^ mice 28 days after angiotensin II infusion. *n* = 13 in vehicle and *n* = 14 in disulfiram groups, respectively. All data are given as mean ± standard deviation. Two-way ANOVA test followed by two group comparison (**A**,**B**) or Student’s *t*-test, ** *p* < 0.01 between two treatment groups.

**Figure 6 biomolecules-13-00899-f006:**
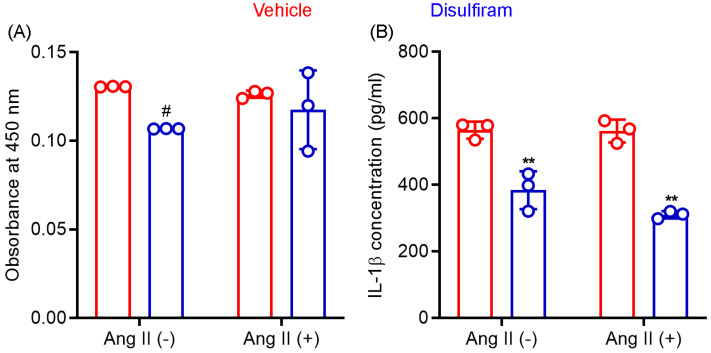
Disulfiram treatment reduces mouse macrophage viability and IL-1β secretion in vitro. Mouse macrophages (cell line RAW264.7) were stimulated with lipopolysaccharide (100 ng/mL) followed by an additional 24-hour incubation in the presence or absence of angiotensin (Ang) II (100 nM) and/or disulfiram (25 mM). (**A**): Cell viability was determined by CCK8 ELISA. (**B**): Supernatant IL-1β concentration was determined by IL-1β ELISA. Two-way ANOVA followed by two group comparison, 0.05 < ^#^ *p* < 0.1 and ** *p* < 0.01 compared to vehicle treatment. *n* = 3.

**Figure 7 biomolecules-13-00899-f007:**
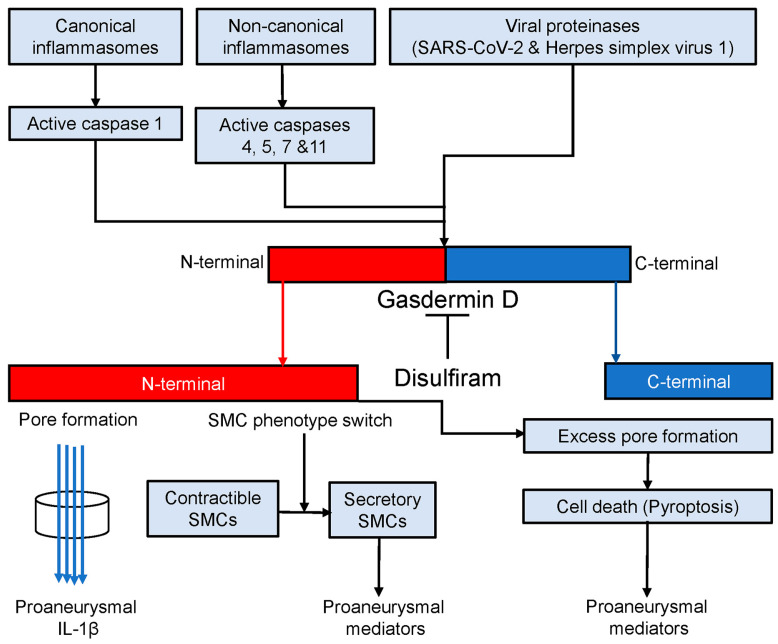
Gasdermin D in AAA pathogenesis. Canonical and noncanonical inflammasome activation and viral proteinases convert procaspases 1, 4, 5, 7, and 11 to corresponding active forms which activates gasdermin D by cleaving it to N- and C-terminals. The C-terminal forms the membrane pores through which proaneurysmal interleukin 1β is secreted. Cell death (pyroptosis) due to excess pore formation also releases various types of intracellular proaneurysmal mediators including cytokines. Alternatively, gasdermin D activation switches smooth muscle cells (SMCs) from contractible towards secretory phenotype leading to proaneurysmal mediator release. Disulfiram blocks active caspase-mediated gasdermin D cleavage and thus suppresses experimental AAAs probably in pore-dependent and -independent manners.

## Data Availability

All data were included within this article.

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
