# Peer review of "Pharmacological Inhibition of Gasdermin D Suppresses Angiotensin II-Induced Experimental Abdominal Aortic Aneurysms"

_biomolecules, 2023, doi:10.3390/biom13060899_

Round 1
Reviewer 1 Report
Manuscript Biomolecules manuscript entitled “Pharmacological inhibition of gasdermin D suppresses angiotensin II-induced experimental abdominal aortic aneurysm” presented very interesting findings. The manuscript was well written for publication in the Biomolecules. However, the manuscript needs to be improved by fixing the following concerns.
There are minor concerns:
1. The authors need to confirm that disulfiram inhibits the cleavage of N-terminal gasdermin D in the AAA aorta;
2. The green fonts in figure 1A and the scale bars in figure 3 were too small to be read;
3. All the figures titles need to be summary sentences;
4. The indicators “vehicle and disulfiram” in figures 1 and 2 were labelled in wrong places;
5. Figure 3 needs to have indication for magnifications;
6. Figure 6 needs to include the verification with human peripheral blood mononuclear cells since the pathophysiological relevance of the findings from Abelson murine leukemia virus-transformed cell line RAW264.7 need to be demonstrated;
7. The authors need to present a new working model by the end of figure set.
Reviewer 2 Report
The present study investigates the effects of gasdermin inhibition by disulfiram on aneurysmal aortic diameter enlargement, AAA incidence, and aneurysm severity in ApoE deficient mice following Ang II infusion. Disulfiram treatment is shown to reduce systemic levels of IL-1beta but not IL-18 and IL-1beta release from LPS-activated macrophages in vitro. The study concludes that gasdermin D inhibition by disulfiram attenuates Ang II-induced experimental AAAs.
As a reviewer, I suggest the following revisions and additions to improve the scientific quality of the manuscript:
- The introduction could be expanded to provide more background information on AAAs, including epidemiology, risk factors, and current treatment options.
- In the second sentence, the term "identified accidently" should be corrected to "identified accidentally."
- The introduction could benefit from a more detailed explanation of the role of inflammation in AAA pathogenesis, as this is the main focus of the manuscript. While the authors briefly mention the role of inflammation in leukocyte accumulation, proteolysis, and angiogenesis, they could expand on these points and provide more specific examples from the literature.
- In line 42, please write mm after 5.5 also, it is not appropriate to use mm only after 4.5 .
- In line 45, the sentence "it necessitates innovating nonsurgical therapeutic options to limit the progression of small clinical AAAs and ultimately reduce mortality due to premature AAA rupture" could be revised for clarity. Specifically, it could be rephrased to clarify that nonsurgical options are needed to address the risk of rupture in patients with small AAAs who do not meet criteria for surgical repair.
- In line 63, it would be helpful to provide specific examples of the vascular disorders in which gasdermin D has been implicated.
- In the discussion, the authors refer to two recent studies that suggest the importance of gasdermin D in AAA pathogenesis. However, they do not discuss the limitations and differences between their study and these studies. In one study, hyperlipidemic mice deficient in gasdermin D were protected against aortic rupture resulting from Ang II infusion. In the other study, smooth muscle cell-specific deletion of gasdermin D attenuated aneurysm diameter expansion and reduced AAA incidence in experimental AAAs. The authors also mention that highly similar to the present study, disulfiram treatment has been shown to reduce aneurysmal diameter, aortic collagen and elastin fragmentation in ApoE deficient mice following Ang II infusion in association with increased protein levels for all NLRP3 inflammasome components. Please discuss the differences between these studies further.
Minor editing of English language required.
